

# Hydrological Controls on Temporal Contributions of Three Nested Forested Subcatchments to DOC Export

Katharina Blaurock[1], Burkhard Beudert[2], Luisa Hopp[1]

[1] Department of Hydrology, Bayreuth Center of Ecology and Environmental Research (BayCEER), University of Bayreuth, 95447 Bayreuth, Germany
[2] Department of Nature Conservation and Research, Bavarian Forest National Park, 94481 Grafenau, Germany

*Correspondence to*: Luisa Hopp (luisa.hopp@uni-bayreuth.de)

**Abstract.** Assessing DOC export from terrestrial systems into inland waters reliably is of paramount importance to understand all processes of the global carbon cycle. Using high-frequency measurements of DOC concentrations via UV-Vis spectrometry, we quantified the DOC export at the outlets of three nested forested subcatchments within a 3.5 km² headwater catchment in the Bavarian Forest National Park, Germany, during a 12 month period. The subcatchments differ with respect to topography, elevation, vegetation and soils. We observed a high flow-weighted DOC export from the entire headwater catchment during spring and autumn. In contrast, during snowmelt, summer and winter, DOC export was low due to low DOC availability and a limited hydrological connectivity, which is the prerequisite for transport processes from terrestrial systems into inland waters. Flow-weighted DOC export also varied between the three subcatchments. Flow-weighted DOC export was always higher in the lower, flat subcatchment than in the upper steep subcatchments, indicating a large DOC store that can be activated, whenever hydrological connectivity is established. This was particularly evident during autumn, when large precipitation events mobilized DOC which had accumulated during the dry summer period and was delivered from fresh leaf litter of deciduous trees. Our data show the strong hydrological control on seasonal DOC export. However, the runoff-based contribution of subcatchments over time is modulated by the interplay of soils, vegetation, topography and microclimate, which can be seen as secondary controls. As hydrological connectivity varies with topography, the relative contribution of topographically different subcatchments varies seasonally. Since climate change is predicted to influence precipitation patterns, spatial and temporal DOC export patterns are likely to change depending on topography.

## 1 Introduction

Organic carbon in inland waters strongly influences the global carbon cycle due to the role it plays for carbon sequestration, transport and mineralization (Battin et al., 2009; Cole et al., 2007). A recent study estimated that the global carbon export from terrestrial systems to inland waters amounts to 5.1 Pg C annually (Drake et al., 2018). Dissolved carbon represents a large part of the exported carbon and therefore influences the net ecosystem balance (Kindler et al., 2011). The majority of the exported dissolved carbon is inorganic (Chaplot and Mutema, 2021) and many uncertainties regarding the contribution of dissolved organic carbon (DOC) to carbon export from terrestrial systems remain. Especially in catchments with wetter and cooler





climates (due to latitude and/or elevation) where organic-rich soils have evolved, DOC is a major component of the exported carbon (Ciais et al., 2008). DOC plays an important role in the context of climate change. It can outgas to the atmosphere from water as the greenhouse gases $CO_2$ and $CH_4$ because of respiration processes. The largest part of all organic and inorganic carbon compounds, which are transported from the terrestrial system to the aquatic system, is emitted to the atmosphere and

not sequestered in the sediments of streams or oceans (Drake et al., 2018; Raymond et al., 2013). But DOC also affects drinking water quality (Sadiq and Rodriguez, 2004; Alarcon-Herrera et al., 1994; Ledesma et al., 2012) and the transport of pollutants (Doerr and Muennich, 1991; Hope et al., 1994; Ravichandran, 2004). Therefore, it is important to characterize the patterns and dynamics of DOC export as accurately as possible.

A meta-analysis of DOC export data found that DOC export from catchments worldwide ranged from 1.2 to over 50 000 kg C

$km^{-2}$ $yr^{-1}$ and that it is negatively related to catchment area and positively to discharge (Alvarez-Cobelas et al., 2012). The main source of DOC in forested catchments is soil organic matter (Batjes, 2014; Borken et al., 2011; Kaiser and Kalbitz, 2012; Kalbitz et al., 2000), and it has been shown that a high proportion of wetlands in a catchment leads to a relatively higher DOC export (Laudon et al., 2011; Zarnetske et al., 2018). Riparian zones are especially regarded to be of importance for DOC export due to the accumulation of soil organic matter and the often high groundwater levels that facilitate transport into the streams

(Ledesma et al., 2018; Mei et al., 2014; Musolff et al., 2018; Ploum et al., 2020; Strohmeier et al., 2013). The largest amounts of DOC are commonly exported during precipitation events (Raymond et al., 2016; Raymond and Saiers, 2010) and DOC concentrations increase with rising discharge (Hobbie and Likens, 1973; Meyer and Tate, 1983). In order to capture the DOC export dynamics precisely, it is necessary to measure DOC concentrations at a high temporal resolution, especially during events. Earlier DOC export estimates have mostly been based on weekly or bi-weekly grab samples leading to a large

uncertainty in export numbers, especially in small catchments (Schleppi et al., 2006a; Schleppi et al., 2006b). By using novel techniques, high-frequency measurements over long periods lead to better estimates of DOC export (Ritson et al., 2022).

DOC export does not only vary with event characteristics but also during and between seasons as a result of varying temperature and wetness conditions (Blaurock et al., 2021; Wen et al., 2020; Werner et al., 2019). Although biogeochemical processes determine the production, quality, and availability of DOC within catchments, water flow processes at catchment

scale eventually control the dynamics and patterns of mobilization and transport of DOC into streams and rivers. Here, hydrological connectivity, influenced by catchment wetness conditions (Detty and McGuire, 2010; McGuire and McDonnell, 2010; Penna et al., 2015), is of paramount importance for solute export (Covino, 2017; Kiewiet et al., 2020). However, patterns and dynamics of hydrological connectivity do not only depend on antecedent precipitation and event characteristics but are also controlled by catchment topography, e. g. shape and slope angle of hillslopes and riparian zones (Detty and McGuire,

2010; Tetzlaff et al., 2014), which, in turn, controls patterns of catchment wetness, i.e., where in a catchment moisture accumulates. Virtual experiments have suggested that the hillslope shape (straight vs. concave) influences the mobilization of nutrients (Weiler and McDonnell, 2006). Therefore, topography is a factor that can be expected to influence DOC mobilization from terrestrial to aquatic systems (Jankowski and Schindler, 2019).





In an earlier study, we showed that the contribution of topographically different subcatchments to event-based DOC export

depended on the establishment of hydrological connectivity between the DOC source areas and the stream (Blaurock et al.,

2021). The objective of this study was to improve our understanding of the temporal and spatial patterns of DOC export from

a forested headwater catchment beyond the event timescale. We conducted high-frequency measurements of DOC

concentrations over one year in the streams of three nested subcatchments to 1) quantify total DOC export from the entire

study catchment in the course of 12 months and 2) explore temporal differences in DOC export among topographically distinct

subcatchments.

## 2 Material and Methods

### 2.1 Subsection

The study was conducted in the Hinterer Schachtenbach catchment ($HS_{tot}$, 3.5 km$^2$), which is part of the Große Ohe catchment

(19.2 km$^2$) in the Bavarian Forest National Park (BFNP, 243 km$^2$), Germany (Figure 1).

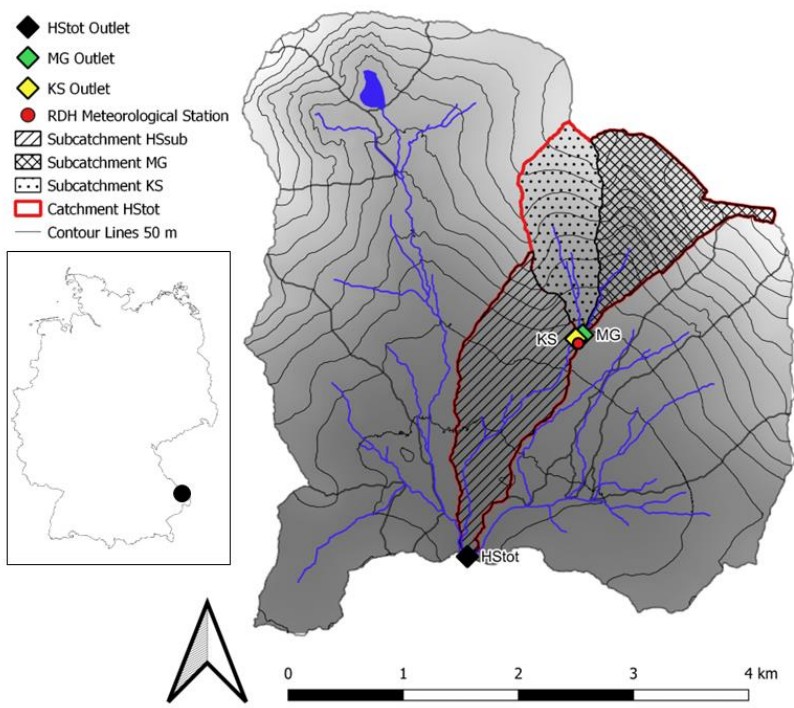


**Figure 1. The studied catchment Hinterer Schachtenbach ($HS_{tot}$) is part of the Große Ohe catchment and located in southeast Germany (black dot). It includes the subcatchments Kaltenbrunner Seige (KS), Markungsgraben (MG) and Hinterer Schachtenbach (HSsub). The high-frequency sampling sites were located at the outlets of the two subcatchments KS (yellow diamond) and MG (green diamond) and at the outlet of the entire catchment HStot (black diamond). Precipitation was measured at the meteorological**

**station RDH (red dot). Both the location of the stream network and the digital elevation model were provided by the Bavarian State Office for Environment. The map of Germany by Vemaps (2022).**





Mean annual precipitation is 1323 mm and mean annual temperature is 6.5 °C (1990 – 2019). Measurements were conducted in three nested subcatchments of $HS_{tot}$ (Table 1): The streams of the upper subcatchments Markungsgraben (MG, 1.1 km$^2$) and Kaltenbrunner Seige (KS, 0.9 km$^2$) join to form the stream of the lower subcatchment Hinterer Schachtenbach ($HS_{sub}$, 1.5 km$^2$). Elevation in the entire catchment ranges from 771 m to 1373 m with a mean slope of 12.0°. MG and KS represent the upper part of the catchment with a mean slope of 15.8° and 14.5° and incised streams, respectively, and $HS_{sub}$ represents the lower part of the catchment with a mean slope of 7.4° and a wide riparian zone. The geology of the Hinterer Schachtenbach catchment ($HS_{tot}$) is dominated by biotite granite (81 %) and cordierite–sillimanite gneiss (14 %). KS and MG are dominated by Cambisols and Podzols with a larger proportion of Cambisols at KS. $HS_{sub}$ is dominated by Cambisols and hydromorphic soils. Almost 40 % of the area of MG is also characterized by rocks, which are interspersed in the soils. Pleistocene solifluction processes created deeper soil layers in the lower catchment parts than in the upper catchment part. The entire catchment is almost entirely covered by forest. Dominant tree species are Norway spruce (Picea abies (L.) Karsten) and European beech (Fagus sylvatica L.). However, large parts of the catchment are in a stage of rejuvenation of mostly Norway spruce due to bark beetle outbreaks in the mid-1990s and 2000s, especially in the subcatchment MG (Beudert et al., 2015). KS is characterized by a higher proportion of mature deciduous forest compared to MG.

**Table 1. Catchment characteristics of the entire catchment $HS_{tot}$ and the subcatchments KS, MG and $HS_{sub}$.**

|  |  | $HS_{tot}$ | KS | MG | $HS_{sub}$ |
|---|---|---|---|---|---|
|  | Area (km$^2$) | 3.5 | 0.9 | 1.1 | 1.5 |
|  | Area ratio of $HS_{tot}$ (%) | 100 | 26 | 31 | 43 |
|  | Elevation (m a.s.l.) | 771 - 1373 | 877 - 1279 | 876 - 1373 | 771 – 1085 |
|  | Mean slope (°) | 12.0 | 14.5 | 15.8 | 7.4 |
| Soils (%) | Cambisol | 66 | 79 | 55 | 65 |
|  | Podzol | 15 | 16 | 34 | 0 |
|  | Hydromorphic soil | 18 | 5 | 5 | 35 |
|  | Other | 1 | 0 | 6 | 0 |
| Vegetation (%) | Rejuvenation | 34 | 28 | 57 | 21 |
|  | Deciduous Forest | 41 | 53 | 29 | 42 |
|  | Coniferous Forest | 9 | 1 | 4 | 17 |
|  | Mixed Forest | 15 | 17 | 8 | 19 |
|  | Other | 1 | 0 | 3 | 1 |

## 2.2 Field measurements

### 2.2.1 Discharge measurements and meteorological data

During the study period from May 30$^{th}$ 2020 to May 29$^{th}$ 2021, the water level was measured at $HS_{tot}$ (Figure 1) every 15 minutes using a pressure transducer (SEBA Hydrometrie GmbH & Co. KG, Kaufbeuren, Germany). Discharge was determined





via tracer dilution (NaCl, TQ-S; Sommer Messtechnik, Koblach, Austria) at 36 occasions to establish a rating curve and

generate 15-min discharge data. Due to a technical failure, no discharge data for $HS_{tot}$ was available from August 1st to

September 3rd, 2020. This data gap was filled using the discharge data of a neighboring catchment within the Große Ohe

Catchment of similar size (Vorderer Schachtenbach, 5.9 km²). Using the discharge values of one year prior to the data gap, a

relationship between the discharge of the two catchments was established ($R^2$=0.94) and calculated values were used to fill the

data gap (Figure S1). For MG, the discharge data for the complete study period were taken from the database of the Bavarian

State Office for Environment (2021) at a 15-min interval. The discharge at KS was measured via tracer dilution (TQ-S; Sommer

Messtechnik, Koblach, Austria) at eight occasions and 15-min discharge values were derived using a relationship between the

highly resolved discharge data of MG and the measured discharge at KS ($R^2$=0.93; Figure S2). Precipitation and temperature

during the study period were measured close to the outlet of MG and KS (meteorological station RDH) and data were provided

by the BFNP. Measuring resolution was one second and storage interval was 10 minutes. Precipitation was measured using a

weighing precipitation gauge (Pluvio² S, OTT Hydromet, Kempten Germany). A ring heater keeps the orifice rim free from

snow and ice. As no long-term temperature data was available for this station, temperature data from a meteorological station

3.8 km east of the catchment outlet (48°55.771' N, 13°27.890' E, 953 m a.s.l.; outside of the map section shown in Fig. 1) was

used to calculate the mean annual temperature.

**2.2.2 High-frequency measurements of DOC concentrations**

DOC concentrations were measured in situ at the outlets of the three subcatchments during the study period using three UV-

Vis (ultraviolet-visible) spectrophotometers (spectro::lyser, s::can GmbH, Vienna, Austria). Device 1 (D1) was installed at the

outlet of $HS_{tot}$ over the whole study period, Device 2 (D2) started at the outlet of MG but had to be replaced by Device 3 (D3)

at the beginning of July 2020 due to a technical failure (after lightning strike). D3 was initially installed at KS and was moved

to MG following the technical failure of D2. After repair, D2 was installed at the outlet of KS in October 2020. This means

that for MG, there is a gap in DOC concentration data for the period from June 23rd to July 8th, 2020, and for KS, there is a

gap in DOC concentration data from July 9th to October 9th, 2020. The spectrometric devices recorded the absorption spectrum

of stream water from 200 to 750 nm with a resolution of 2.5 nm every 15 min. DOC concentrations were quantified using the

internal calibration based on the absorption values using the software ana::pro. In order to refine the internal calibration, the

DOC concentrations measured by the three UV-Vis spectrophotometers were corrected using grab stream samples taken over

the course of the study period at various discharge conditions ($n_{D1}$ = 52, $n_{D2}$= 22, $n_{D3}$ = 44; Figure S3-S5). Samples were

filtered in the field using polyethersulfone membrane filters (0.45 μm) or polycarbonate track etched membrane filters (0.45

μm). All samples were stored until further analysis at 4°C. DOC concentrations of the grab samples were analyzed by thermo-

catalytic oxidation (TOCL analyzer; Shimadzu, Kyoto, Japan). For further analysis, the corrected values ($R^2$ for D1 = 0.98, $R^2$

for D2 = 0.87 and $R^2$ for D3 = 0.97) were used. As no drift of the DOC concentration could be identified in the measured

signal, we decided against a correction for biofouling as done by Werner et al. (2019). However, the sensor optics were

manually cleaned in the field every 2 weeks using cotton swabs.



### 2.2.2 Data analysis

For the evaluation of the temporal DOC dynamics during our study period, we defined "hydrological periods" based on discharge behavior and changes in precipitation patterns. Since these hydrological periods overlap with calendrical seasons to a larger extent, we designate the hydrological periods by the commonly used names for calendrical seasons (Table 2). We prefer this definition of hydrological periods for evaluating DOC export over the usage of calendrical or meteorological seasons to avoid the arbitrary attribution of precipitation and runoff events. Spring started with the end of snowmelt and ended with the beginning of a dry period in July 2020, which was defined as summer. Summer ended with the end of a long dry period prior to the beginning of a rainy period, which was defined as autumn. Winter started after the last precipitation event that did not fall as snow and when the temperature fell below zero. Snowmelt was defined as the period when temperatures rose above zero, a snow cover was present and snowmelt was clearly visible in discharge data at HS. The length of the hydrological periods is presented in days. However, these are rounded values as the start or end of a hydrological period did not automatically fall on the start or end of a day. For the calculation of mean daily DOC export, non-rounded values were used. All hydrological periods were defined using discharge data of $HS_{tot}$ and precipitation data from the meteorological station RDH. Runoff ratios were calculated as the ratio of cumulative area-normalized discharge to the amount of precipitation from RDH during each hydrological period. Data analysis was performed using MATLAB (The MathWorks, Inc.).

**Table 2. Hydrological periods and number of days with measured DOC data for $HS_{tot}$ and for all subcatchments. Due to the technical failure of device D2, there is no period with data for all three subcatchments available during summer.**

|  | Dates | Total number of days | Number of days with DOC data for $HS_{tot}$ | Number of days with DOC data for all subcatchments |
|---|---|---|---|---|
| Spring | 30.05.2020 – 30.06.2020<br>02.05.2021 – 29.05.2021 | 60 | 60 | 52 |
| Summer | 01.07.2020 – 24.09.2020 | 87 | 83 | 0 |
| Autumn | 25.09.2020 – 21.11.2020 | 57 | 57 | 34 |
| Winter | 22.11.2020 – 21.12.2020<br>30.12.2020 – 28.01.2021 | 60 | 60 | 58 |
| Snowmelt | 22.12.2020 – 29.12.2020<br>29.01.2021 – 01.05.2021 | 101 | 91 | 87 |

DOC export from $HS_{tot}$ and subcatchments was calculated as follows for the entire study period and the hydrological periods:

$$DOC \ export = \sum_{i=1}^{n} c_i \times Q_i \times 15$$





where $c_i$ is the DOC concentration [mg L$^{-1}$] and $Q_i$ is the discharge [L min$^{-1}$] for n 15-minute intervals. Small data gaps of several hours were interpolated linearly. For the year investigated, DOC data for HS$_{tot}$ was available for 351 days out of 365
days. Daily DOC export averages were calculated by dividing the DOC export measured during one hydrological period by the length of the respective hydrological period. Total DOC export from HS$_{tot}$ for each hydrological period was calculated as the sum of the measured DOC export and, for each day without data, the calculated daily average (for the corresponding hydrological period). Results from all subcatchments were available on 231 days during the study period. Flow-weighted DOC export for a period was calculated by dividing the measured and estimated DOC export during this period by the
cumulated discharge generated in this period. When comparing the contribution of different subcatchments to total DOC export, we used the ratio of the subcatchment area to the total catchment area as a benchmark (KS/HS$_{tot}$ = 0.26, MG/HS$_{tot}$ = 0.31, HS$_{sub}$/HS$_{tot}$ = 0.43). All results presented for HS$_{sub}$ were derived from the difference of HS$_{tot}$ and the upper subcatchments (MG and KS). For these calculations, we assumed that all DOC entering HS$_{sub}$ was still present at the outlet of HS$_{tot}$ and did not leave the stream by surface water-groundwater exchange or in-stream processes on its passage towards the catchment
outlet.

## 3 Results

### 3.1 DOC concentrations, discharge and precipitation during the investigation period

With 1280 mm of precipitation at RDH, the investigated period was drier than the long-term annual average (1535 mm for RDH 1990 – 2019). The mean temperature during the study period was higher (6.8°C) than the long-term annual average
(6.5°C, 1990 – 2019).

The spring period was characterized by frequent precipitation events. During the summer period, long dry periods were interrupted by a few precipitation events. In autumn, several large precipitation events occurred, leading to sharply rising DOC concentrations up to 20 mg/L (Fig. 2). During the winter period, precipitation fell as snow, which led to low discharge and low DOC concentrations. Winter was interrupted by one large snowmelt event induced by rising temperatures and rainfall towards
the end of December. During snowmelt, diurnal fluctuations of discharge and DOC concentrations were observed following the amplitudes of air temperature. The highest discharge value measured during the study period reached 0.75 m$^3$ s$^{-1}$ at the onset of the snowmelt period. Additionally, some rain events enhanced snowmelt. DOC concentrations clearly increased with discharge during precipitation events (Fig. 2), observed during events in the summer and the autumn period and during the short snowmelt event in late December. During baseflow, DOC concentrations usually vary between 2 and 3 mg L$^{-1}$ at MG
and HS$_{tot}$ (Da Silva et al., 2021).





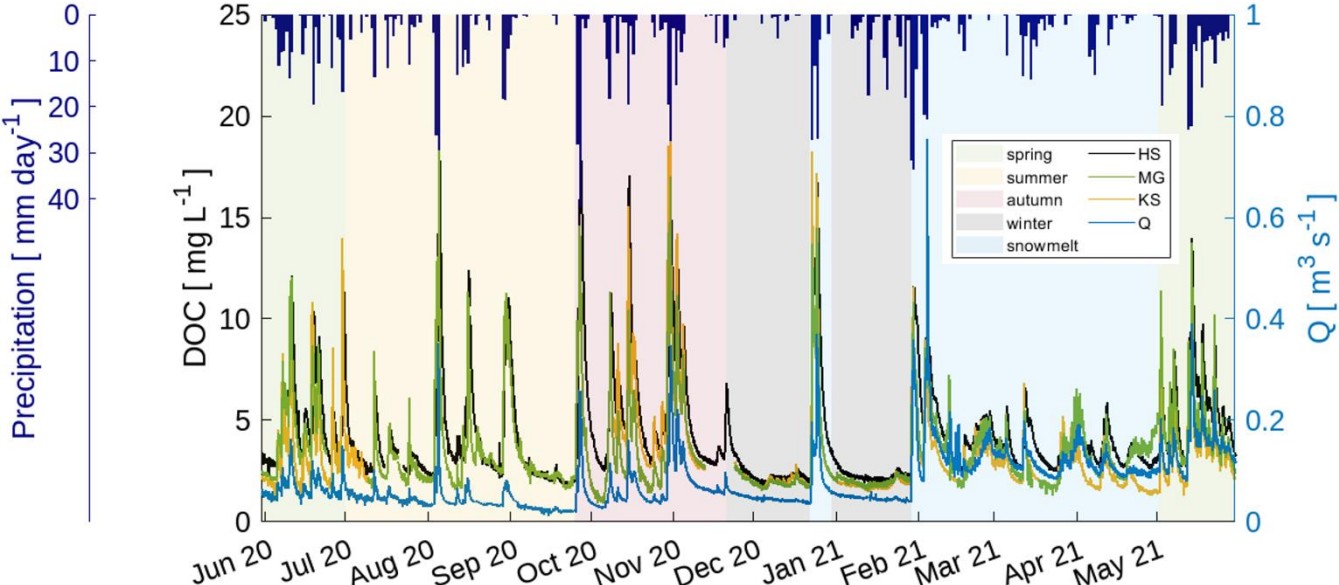

**Figure 2: DOC concentrations at the outlets of the subcatchments KS (yellow), MG (green) and HS (black) and discharge (Q) at HS$_{tot}$ (blue). The shaded areas correspond to the delineated hydrological periods.**

**3.2 DOC export from the entire catchment HS$_{tot}$**

The annual DOC export of HS$_{tot}$ was 13760 kg or 3931 kg km$^{-2}$, if including both the measured (13136 kg, 351 days) and the estimated (624 kg, 14 days) data (Table 2). The largest amounts of DOC were exported during snowmelt (40.5 %), spring (24.6 %) and autumn (20.3 %), whereas the export in summer (10.4 %) and winter (4.2 %) was much lower. In the following, all results refer only to periods with measured data. The highest mean daily DOC export was observed during the hydrological

periods of spring and snowmelt, followed by autumn (>45 kg d$^{-1}$), whereas winter and summer exhibited markedly lower mean daily DOC release (< 20 kg d$^{-1}$) (Table 3).

**Table 3. Total precipitation P$_{tot}$ [mm] at RDH, total DOC export [kg], number of days per hydrological period, and mean daily DOC export [kg d$^{-1}$] at HS$_{tot}$ during the periods for which measured data was available (see Table 2).**

|  | P$_{tot}$ [mm] | Total DOC export [kg] | Days [d] | Mean daily DOC export [kg d$^{-1}$] |
|---|---|---|---|---|
| Spring | 327 | 3381 | 60 | 56.4 |
| Summer | 214 | 1378 | 83 | 16.7 |
| Autumn | 262 | 2794 | 57 | 48.8 |
| Winter | 129 | 576 | 60 | 9.5 |
| Snowmelt | 332 | 5007 | 91 | 55.1 |
| All periods | 1264 | 13136 | 351 | 37.4 |






As the hydrological periods had different lengths (Table 3) and DOC and discharge are positively related (Fig. 2), we also examined the flow-weighted DOC export normalized by the length of the hydrological periods to remove the influence of discharge (Fig. 3). Autumn proved to be the hydrological period with the highest daily flow-weighted DOC release, followed by the spring period. Although hydrologically being quite different, the summer, winter and snowmelt periods exhibited similar 205 mean daily flow-weighted DOC export.

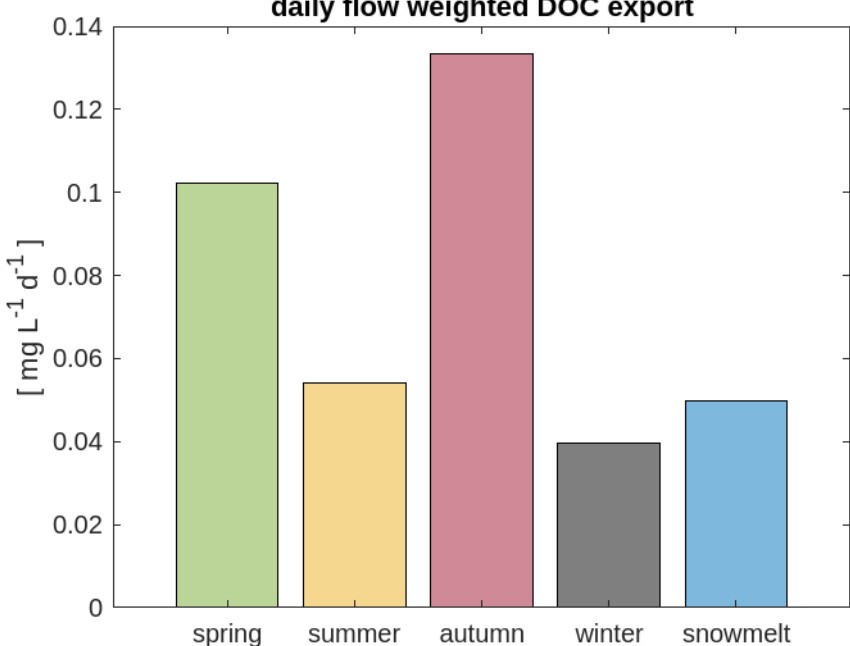

**Figure 3. Mean daily flow-weighted DOC export from the Hinterer Schachtenbach catchment (HS$_{tot}$) during the different hydrological periods.**

Plotting discharge and DOC concentration time series as cumulative curves illustrated the DOC export behavior more clearly (Fig. 4). Cumulative discharge increased almost linearly during the study period with a break in slope with the onset of snowmelt at the end of January 2021, indicating an increased runoff generation during the snowmelt and spring period that lasted until the end of the study period.

In contrast, pronounced step-like increases were visible in the cumulative DOC export curve, particularly in autumn and during 215 early snowmelt, emphasizing the relevance of runoff events for DOC release within a short time.

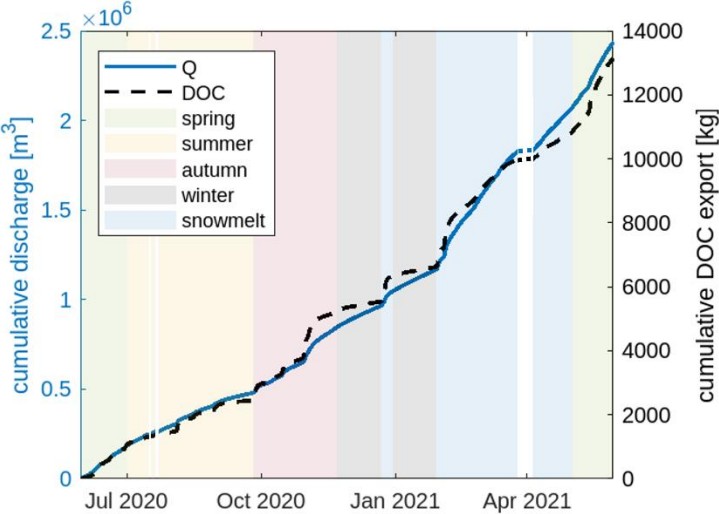

**Figure 4. Cumulative discharge (blue line) and cumulative DOC export (black dashed line) from HS$_{tot}$ during the different hydrological periods (shaded background). Missing data (white background) are not accounted for.**

### 3.2 DOC export behavior of the subcatchments

The area-normalized cumulative export of the three subcatchments showed a similar pattern running parallel from late spring until the middle of the autumn period, when HS$_{sub}$ began to export less DOC than MG and KS (Fig. 5). The cumulative export of HS$_{sub}$ remained smaller than the export of MG and KS throughout the winter period. In winter, DOC export was generally low. Export of area-normalized DOC from all three subcatchments during the autumn and the winter period occurred in a step-like pattern whereas with the onset of snowmelt, DOC export happened more continuously over time, with particularly

strong increases in HS$_{sub}$ and MG at the beginning of the snowmelt period. Strong increases of DOC export could also be seen in spring following precipitation events. At the end of the study period, MG showed the highest DOC export per unit area, whereas the area-normalized export of KS was lower by 32%.





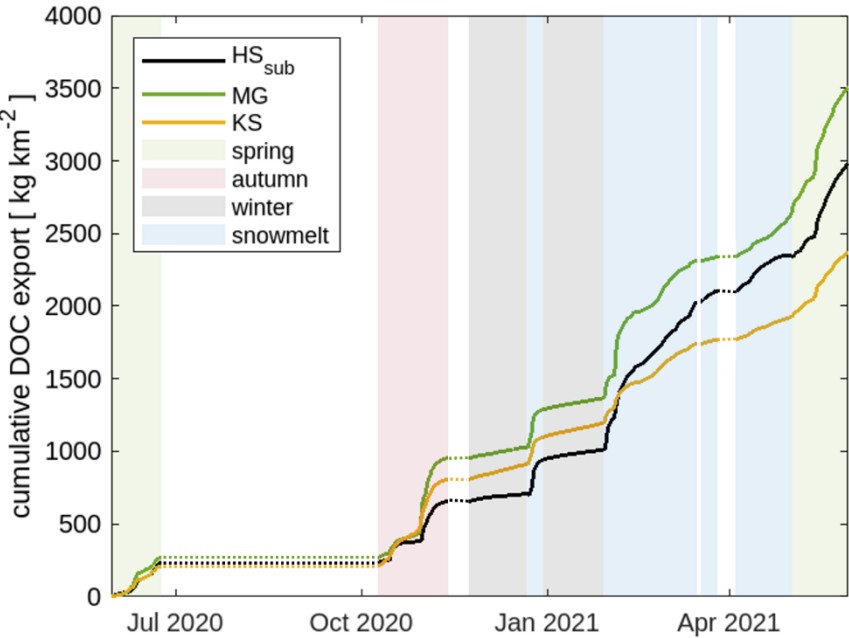

**Figure 5. Cumulative DOC export per unit area from the subcatchments MG, KS and HS_sub during the different hydrological periods**
**(shaded background). Note: For periods when not all three sensors were working (white background), cumulative DOC export from**
**the subcatchments was not calculated.**

Area-normalized daily DOC export was high during snowmelt, spring and autumn and low during winter (Fig. 6a). Flow-weighted DOC export was always highest in the $HS_{sub}$ catchment, particularly during autumn, underlining the relevance of $HS_{sub}$ for DOC export (Fig. 6b). However, the relative contribution of the subcatchments to total DOC export varied between the seasons (Fig. 6c).



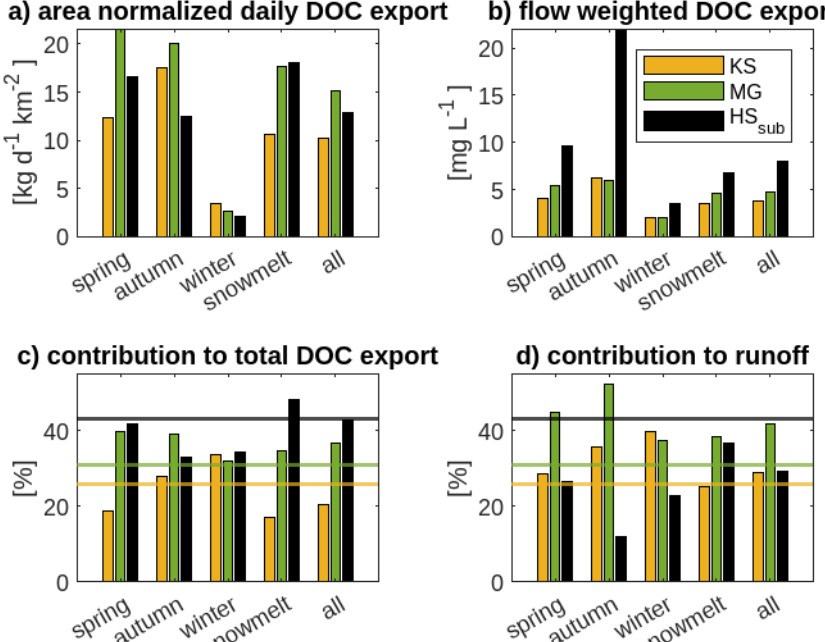

**Figure 6. Contributions to DOC export and runoff of the subcatchments KS, MG and HS$_{sub}$ for spring, autumn, winter , snowmelt and for the sum of those hydrological periods (all): a) area-normalized daily mean DOC export in kg d$^{-1}$ km$^{-2}$, b)  DOC export weighted by discharge generated in each subcatchment in mg L$^{-1}$, c) contribution to total DOC export in %, d) contribution to total runoff in %. For c) and d), the horizontal lines correspond to the value expected according to the area ratio (see Table 1).**

During snowmelt and spring, the order of contribution to total DOC export was as expected in terms of catchment area as HS$_{sub}$ contributed most, followed by MG and KS. In winter, all subcatchments contributed equally to total DOC export due to a larger contribution than expected of KS and a smaller contribution than expected of HS$_{sub}$. In autumn, MG contributed most to total DOC export, whereas HS$_{sub}$ contributed less than during the other hydrological periods (Fig. 6c). The contribution of MG was larger than expected during the entire year and even exceeded the contribution of the larger subcatchment HS$_{sub}$ in autumn. MG also contributed more to runoff than expected during all hydrological periods, whereas HS$_{sub}$ contributed less than expected to runoff (Fig. 6d). KS contributed more to runoff than expected from proportional area in autumn and winter. The runoff ratios were highest for MG during all hydrological periods except for winter and summer, when KS showed the highest runoff ratio (Table 4). The runoff ratios were lowest for HS$_{sub}$ during all hydrological periods. The highest runoff ratios could be found during snowmelt for all subcatchments. The lowest runoff ratio could be found in spring for KS, in summer for MG and HS$_{tot}$ and in autumn for HS$_{sub}$. Overall, HS$_{tot}$ exhibited a mean runoff coefficient of 0.57 for the entire study period, with runoff coefficients ranging between 0.38 and 0.94.

**Table 4. Runoff ratios for the entire catchment HS$_{tot}$ and the subcatchments KS, MG and HS$_{sub}$ during the periods for which measured discharge data was available for all subcatchments.**

|  | HS$_{tot}$ | KS | MG | HS$_{sub}$ |
|---|---|---|---|---|
| Spring | 0.48 | 0.54 | 0.68 | 0.30 |
| Summer | 0.38 | 0.62 | 0.47 | 0.18 |
| Autumn | 0.40 | 0.56 | 0.63 | 0.13 |
| Winter | 0.53 | 0.82 | 0.64 | 0.29 |
| Snowmelt | 0.94 | 0.92 | 1.15 | 0.81 |

The contribution to DOC export and the contribution to runoff of the subcatchments differed (Fig. 6 and Table 5). The

contribution to DOC export from KS and MG during the entire period was lower than their contribution to runoff. On the

contrary, the contribution of HS$_{sub}$ to DOC export was markedly larger than its contribution to runoff for all hydrological

periods (Table 5).

**Table 5. Ratio between relative contribution to DOC and relative contribution to runoff generation for the three subcatchments**
**during the hydrological periods (without summer)\*.**

|  | KS | MG | HS$_{sub}$ |
|---|---|---|---|
| Spring | 0.66 | 0.88 | 1.57 |
| Autumn | 0.79 | 0.74 | 2.77 |
| Winter | 0.85 | 0.86 | 1.51 |
| Snowmelt | 0.68 | 0.9 | 1.32 |

\*If the ratio is larger than 1, then the contribution to DOC release is
relatively larger than the contribution to runoff generation.

## 4 Discussion

### 4.1 Variations of DOC export from the entire headwater catchment over time

The annual DOC export of the catchment of 3,931 kg km$^{-2}$ is in the range of other European forested catchments (Ågren et al.,

2007; Tittel et al., 2013) with values, e.g., being higher than measured in a Mediterranean catchment (Bernal and Sabater,

2012), but lower than in a wetland-dominated catchment (Strohmeier et al., 2013). DOC export of the entire Große Ohe

catchment (which the Hinterer Schachtenbach catchment is a subcatchment of) was estimated to be very similar as well with

4,200 kg km$^{-2}$ (Beudert et al., 2012). In general, it has been found that precipitation, and with that discharge, is a primary

control for DOC export (Wei et al., 2021) and that wetter years result in a higher export of DOC than drier years (Jager et al.,

2009).

The absolute export of DOC during the hydrological periods was highest during snowmelt and lowest during the winter period.

However, these DOC amounts are affected by the lengths of the hydrological periods and by the runoff generated. Since



discharge and DOC concentrations are positively correlated and discharge is the main driver for DOC export, the flow-weighted evaluation of DOC export is necessary to disentangle the relevance of factors other than discharge, such as topography, soils or vegetation, for the release of DOC into streams. The flow-weighted values can also be interpreted as the

efficiency of a catchment to mobilize DOC per unit generated discharge.

When evaluating the flow-weighted export of DOC from the headwater catchment Hinterer Schachtenbach (Fig. 3), we observed that DOC export was low during the periods of summer, winter, and snowmelt, compared with the other two hydrological periods of spring and autumn. In summer, discharge was at baseflow with occasional short increases in discharge due to precipitation events. During these events, DOC concentrations rose up to 20 mg L$^{-1}$, reaching the highest concentrations

observed during the study period. This indicated that DOC was available and ready to be mobilized. However, since discharge was mostly low, the exported DOC load remained overall also low for the summer period. Less precipitation input and an increased water demand by vegetation led to decreasing catchment wetness and drier soils. We argue that hydrological connectivity between the DOC source areas and the stream was established less often and to a lesser spatial extent leading to the inhibition of DOC export as explained by Blaurock et al. (2021), derived from an analysis of four precipitation events. This

analysis suggested that the lower flatter part of the headwater catchment with the wide riparian zone may be affected more often by missing hydrological connectivity. At the same time, previous work indicated that this catchment part can release substantial amounts of DOC that accumulates in microtopographic depressions into the stream once hydrological connectivity is established (Blaurock et al., 2022). Overall, this again points to DOC export being transport-limited in this catchment, as has been observed in many other catchments (Zarnetske et al., 2018). The relevance of this transport-limitation of DOC release

into streams may increase in the future if summer droughts will occur more frequently with climate change.

During the winter period, with precipitation falling as snow and adding to the snowpack, discharge at the outlet of the Hinterer Schachtenbach was characterized by a constant low level, except for a brief snowmelt period following rising air temperatures and a rain event at the end of December 2020. Also, DOC remained at concentrations typical for baseflow conditions in this catchment (Da Silva et al., 2021), resulting in a low DOC release from the catchment. DOC production is generally low during

winter because of cold temperatures (Kalbitz et al., 2000) and snowfall. DOC produced during autumn accumulates in the soil and can be readily mobilized, as evidenced by the strong increase in DOC concentrations during the short snowmelt event at the end of December 2020.

The snowmelt period led to higher discharge and an increase of DOC concentrations in streamflow. Several studies have shown that DOC export during snowmelt can be substantial (Jager et al., 2009; Seybold et al., 2019; Wilson et al., 2013). The diurnal

fluctuations of discharge and DOC during snowmelt indicated that DOC export was strongly controlled by hydrological processes, which responded to the freeze-thaw cycle of the snowpack. The largest DOC export could be observed at the beginning of the snowmelt periods probably due to the flushing of topsoil DOC that had accumulated during winter (Ågren et al., 2010; Boyer et al., 1997) as a result of the leaching of DOC from the autumnal litter input or DOC production in snow-covered soil (Brooks et al., 1999). However, neither discharge nor DOC concentrations showed any pronounced event

responses, which again led to overall low flow-weighted DOC export (Fig. 3) and a more continuous release of DOC (Fig. 4).





During the spring period at the beginning and at the end of our study period, the daily flow-weighted DOC almost doubled, compared to the summer, winter and snowmelt periods (Fig. 3). Soils were saturated after the snowmelt period and, therefore, hydrological connectivity between the DOC sources and the stream existed facilitating a high continuous DOC export (Fig. 4) and the connection of distal DOC sources (Croghan et al., 2023) during the spring events. Furthermore, the increasing

temperatures during spring had enhanced DOC production, thereby increasing the potential mobilizable DOC pool.

The autumn period showed the highest flow-weighted DOC release. The onset of larger precipitation events led to sharp increases in discharge and DOC concentrations between 15 and 20 mg L$^{-1}$. In autumn, DOC export was clearly driven by the large precipitation events. Moreover, it is possible that DOC had accumulated in the soils during summer due to 1) the warm temperatures, which enhance biological activity, and 2) the lack of precipitation events, therefore limiting DOC flushing

(Dawson et al., 2008; Kawasaki et al., 2005; Seybold et al., 2019; Wei et al., 2021). In autumn, leaf litter from deciduous trees may constitute an additional important DOC source potentially amplifying DOC export during this period (Hongve, 1999; Meyer et al., 1998). Long-term observations of phenology and litter fall in the Bavarian Forest National Park show that over 80 % of litter fall is likely to occur during the here defined autumn period at all elevations (unpublished data, Bavarian Forest National Park Administration, 2022). It has also been shown that leaf litter leads to an annual carbon input of 150,000 kg

C/km$^2$ in a neighboring catchment (Forellenbach) and can therefore substantially contribute to DOC export (unpublished data, Bavarian Forest National Park Administration, 2022).

To conclude, we saw pronounced temporal variations in DOC export from the Hinterer Schachtenbach catchment during our one-year study period. As has been observed elsewhere, DOC showed a chemodynamic behavior with a positive relation to discharge. Hence, the most important factor for DOC export was precipitation and, with that, discharge. Precipitation and

subsequent runoff events contributed substantially to DOC release within short times. After normalizing DOC export with runoff, the hydrological period of autumn, after a dry summer period, showed the highest DOC release, caused by the often-missing hydrological connectivity in the preceding summer period and a subsequent accumulation of DOC in the soils of the catchment. The hydrological periods of summer, winter and snowmelt exhibited low flow-weighted DOC export, driven by the interplay between runoff generated and available DOC pool that can be mobilized. The DOC production in a catchment,

in turn, is influenced by temperature and the organic matter from vegetation available for decomposition.

### 4.2 The relative importance of the subcatchments for runoff generation and DOC export

The one-year dataset of 15-min DOC concentrations and discharge values at three topographic positions within the Hintere Schachtenbach catchment illustrated the relative role of the subcatchments for runoff generation and the mobilization of DOC. The area-normalized DOC release of the subcatchments (Fig. 6a) changed over time and also in relation to each other. Snowfall

and low temperatures led to a low DOC export in winter. The relatively high contribution of KS during autumn and winter could be due to the influence of litter fall on DOC export as KS is the subcatchment with the highest fraction of deciduous trees (Table 1). The breakdown of this additional input of organic matter could lead to a constant DOC production even during the winter months (Hongve, 1999). Additionally, litter fall could induce a lower water loss via canopy interception in autumn





leading to more infiltration into the forest soils and active flow pathways, as indicated by the high runoff ratio at KS during

winter (Table 4). After the snowmelt period, the relative importance of MG for DOC export increased. This could be a consequence of a delayed snowmelt. The subcatchment MG extends up to 1373 m a.s.l., which is 100 m higher than the neighboring subcatchment KS, leading to colder temperatures and a longer lasting snowpack (unpublished data, Bavarian Forest National Park Administration, 2022). Therefore, a delayed snowmelt at MG could contribute to the higher DOC export in the spring period. Similar observations were made by Boyer et al. (1997) who found a delayed contribution of upslope

catchment parts to the total DOC export due to an asynchronous melting of the snow in an alpine catchment.

Moreover, the contribution of MG to runoff from the entire Hintere Schachtenbach catchment was markedly higher than expected from the area proportion during all hydrological periods and except for the winter period always the highest among the three subcatchments (Fig. 6d). One reason for this could be the higher altitude of the subcatchment MG, resulting in a higher precipitation due to orographic effects. Observations in the Große Ohe catchment have shown that precipitation

increases by around 100 mm per 100 m increase in altitude. MG is characterized by steep slopes with over 22 % of the area having slopes with 21 – 25° (in comparison to 2 % in the neighboring subcatchment KS). MG possesses also a higher percentage of Podzol soils that usually have a higher fraction of sand and subsequently large pores, leading to a higher hydraulic conductivity (Scheffer-Schachtschabel, 2018). The permeable soils in MG are also interspersed with large rocks, facilitating the occurrence of preferential flow paths. These factors may result in a quick movement of infiltrated water towards the stream

channel and a high runoff generation. In contrast, at KS, interspersed rocks are much less prominent leading to slower flow processes. These differences in topography as well as soils could explain the differences in runoff generation between the two upslope subcatchments KS and MG.

The lower subcatchment HS$_{sub}$ exhibited particularly low mobilized DOC loads per unit area in the autumn period (Fig. 6a). This period was also the one with the lowest runoff contribution of HS$_{sub}$ (Fig. 6d). In our previous event-based study, we had

identified the missing hydrologic connectivity in the subcatchment HS$_{sub}$ after summer drought as a dominant reason for inhibited DOC mobilization from this subcatchment towards the stream during autumn (Blaurock et al., 2021). The flat riparian zone of HS$_{sub}$ is dependent on a high level of catchment wetness to enable hydrological connectivity in the shallow subsurface, which is the prerequisite for DOC export, whereas the steeper upper subcatchments are able to maintain flow towards the stream even during dry conditions. The one-year dataset confirmed our event-based interpretation.

However, considering only the area-normalized export of DOC may skew our view of catchment functioning. Normalizing the DOC export values with the cumulative discharge that was generated in each subcatchment per hydrological period allows to consider mobilization factors other than discharge. HS$_{sub}$ always showed the highest flow-weighted DOC export (Fig. 6b), e.g., in HS$_{sub}$, one unit discharge mobilized around four times as much DOC as in the other subcatchments during the autumn period. This suggests that the lower subcatchment possesses a large DOC store that can be activated whenever hydrological

connectivity is established. In our previous work at the study site, we identified microtopographic depressions ("pools") in the riparian zone that fill regularly with water from the bottom during large events and periods of high catchment wetness as hot spots for DOC accumulation and DOC release towards the stream (Blaurock et al., 2022). In addition, the lower elevation of





HS$_{sub}$ and resulting differences in mean air and soil temperatures may contribute to an overall higher production and availability of DOC. Particularly in autumn, after the dry summer, DOC had accumulated and was readily exported into the streams during

the first large rainfall events, leading to the pronounced flow-weighted DOC export from HS$_{sub}$.

Comparing the daily flow-weighted DOC export per hydrological period from the entire catchment HS$_{tot}$ (Fig. 3) and the subcatchments (Fig. S6) revealed that each subcatchment behaved as the entire catchment Hinterer Schachtenbach but that the differences between the hydrological periods varied for the three subcatchments (note: summer could not be evaluated). The difference between spring and autumn was particularly pronounced for HS$_{sub}$ and only small for MG. As stated above, this

could be explained with the missing hydrological connectivity between DOC source areas and the stream during the dry summer that led to an accumulation of DOC in the soils of HS$_{sub}$. In the transition from winter to snowmelt, all subcatchments seemed to contribute equally to the doubling of flow-weighted DOC export of HS$_{tot}$. As DOC export was limited during winter, DOC which had accumulated in the topsoil during the winter months could now be flushed by meltwater and precipitation events (Ågren et al., 2010; Boyer et al., 1997). From snowmelt to spring, the strongest change was observed in HS$_{sub}$,

suggesting that HS$_{sub}$ with its large available DOC pool was most likely driving the increase in flow-weighted DOC export.

The upper subcatchments mobilized substantial amounts of DOC because of their high runoff generation, as indicated by their runoff ratios that were always higher than for the lower subcatchment (Table 4). At the same time, their contribution to DOC export was not as high as their contribution to runoff within the Hintere Schachtenbach catchment (Table 5). In contrast, despite the often low runoff ratio of the lower subcatchment HS$_{sub}$ (except for the snowmelt period), it contributed disproportionately

to DOC export in relation to runoff, illustrating the DOC source strength of this catchment part.

Analyzing area-normalized as well as flow-weighted DOC export, in combination with the relative contributions of the subcatchments to DOC export and runoff generation of the Hinterer Schachtenbach catchment, revealed different factors that drive the release of DOC. The upper subcatchment MG contributed substantially to DOC export because of the pronounced runoff generation taking place in MG. Topography, soils and vegetation can explain the differences in runoff generation and

DOC release between the two steep upslope subcatchments MG and KS. The lower subcatchment HS$_{sub}$ was limited in its contribution to DOC export during times of low catchment wetness since it is highly dependent on the establishment of hydrological connectivity between the DOC source areas in the flat riparian zone and the stream. However, due to its deeper soils and microtopographic features that act as DOC production and accumulation hot spots, high amounts of DOC are present that can be readily mobilized, whenever wetter hydrological conditions occur and hydrological connectivity is established.


The transport-limited nature of DOC mobilization has been shown before (Zarnetske et al., 2018). Our study demonstrates that hydrological processes strongly control the export of DOC but that the runoff-based contribution of subcatchments over time is modulated by the interplay of soils, vegetation, topography and microclimate, which can be seen as secondary controls.





## 5 Conclusions

Our high-resolution data show that DOC export varies seasonally as a result of both varying DOC production and varying hydrological connectivity, which is controlled by topographical position. Topography, together with soil hydraulic properties and vegetation (differences in litter fall and dynamics of interception), governs the accumulation of water in the landscape, the formation of hydrological flow pathways and the establishment of hydrological connectivity towards the stream channel. These hydrological processes control the DOC export into streams and lakes. Although hydrological connectivity is a general state

that needs to be present to enable the transport of solutes towards the stream, some catchments are particularly sensitive towards the establishment of hydrological connectivity, due to their combination of topographic, soil and vegetation characteristics. Different catchment parts can mobilize high DOC loads for different reasons, e.g., due to strong runoff generation and/or the presence of DOC production and accumulation hot spots that are activated upon occurrence of hydrological connectivity towards the stream. In next decades, temperate ecoregions are likely to be influenced strongly by climate change as a result of

rising temperatures and changing precipitation patterns (Intergovernmental Panel on Climate Change, 2021), influencing terrestrial-aquatic linkages within the carbon cycle with potentially large consequences for DOC export (Wit et al., 2021). Rising temperatures could increase DOC production and, therefore, the potential DOC pool. Higher temperatures in winter would reduce the amount of precipitation that falls as snow. Consequently, more DOC could be exported during winter and the flushing effect during snowmelt would be reduced.  DOC export could be distributed more evenly during the winter and

spring months. However, more extreme precipitation events and prolonged drought periods could lead to a stronger contrast in DOC export patterns during summer and autumn. Drought periods would also limit DOC production and a low DOC export in summer and autumn could then lead to a higher DOC export during the following winter and snowmelt (Ågren et al., 2010). Additionally, topography and vegetation seem to influence the contribution of the different catchment parts to DOC export throughout the year. Drought periods could limit DOC export from the lower catchment parts, which are dependent on

hydrological connectivity, leading to an increasing importance of the upper catchment parts for DOC export. This effect could be enhanced by longer warm periods at higher altitudes.

Our study highlights that we need a thorough understanding of catchment hydrological processes to understand the patterns and dynamics of carbon export from terrestrial systems into inland waters (Catalán et al., 2016). High-resolution data is essential to capture the highly dynamic DOC export behavior and to include fast mobilization processes in DOC export

calculations. Therefore, high-resolution data contributes substantially to a better understanding of the seasonally varying contribution of landscape parts to streamflow generation and associated DOC mobilization and export.

*Data availability*. The data set used in this study is available at Figshare via https://doi.org/10.6084/m9.figshare.22770365. During the peer-review process, we kindly ask the reviewers to use the following link to access the dataset:

https://figshare.com/s/1216d7fecbd3f0488d83. The official DOI will be activated upon publication. For data analysis, MATLAB (The MathWorks, Inc.) was used.





*Supplement*. The supplement related to this article is available online at:

*Author contributions*. Conceptualization by LH, KB and BB; investigation, methodology and formal analysis by KB; Writing – original draft preparation by KB; Writing – review & editing by LH and KB.

*Competing interests*. The contact author has declared that none of the authors has any competing interests.

Acknowledgements. This research was funded by the Rudolf and Helene Glaser Foundation and the Zempelin Foundation of the Stifterverband für die Deutsche Wissenschaft, as part of the project "Influence of natural factors on concentration, quality and impact of dissolved organic carbon in the Bavarian Forest National Park" (Project No. T0083\30771\2017\kg). The authors would like to thank the Bavarian Forest National Park (BFNP) administration for providing physiographic and meteorological data, and the BFNP staff for their helpful assistance with the installation and maintenance of the field equipment. Funded by
the Open Access Publishing Fund of the University of Bayreuth.

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
