# Peer review of "Hydrological Controls on Temporal Contributions of Three Nested Forested Subcatchments to the Export of Dissolved Organic Carbon"

_Hydrology and Earth System Sciences, 2024_

## Author Comment (AC1)

**Authors' response to Comments of Reviewer 2**

**Review of hess-2024-250**

In their manuscript, entitled "Hydrological Controls on Temporal Contributions of Three Nested Forested Subcatchments to DOC Export", Blaurock et al. analyzed DOC export across different hydrological periods. By the use of high-frequency data, they could show clear differences between the catchments in terms of the timing and magnitude of DOC export, which the authors could explain by differences in soil, topography, vegetation, and microclimate. The manuscript is well-written and contributes interesting insights into the drivers of riverine DOC export at the headwater catchment scale. My comments might be abundant, but they are largely of minor character. I do agree with the main points raised by reviewer 1. Furthermore, I did not agree with the author's repeated argumentation that the most important factor for DOC export was precipitation and, with that, discharge (see my detailed comment to lines 272-274). Furthermore, the authors should be careful in stating that certain catchments characteristic "control" DOC export, while not having directly tested this, but "only" having provided a sound explanation. Overall, studies like this one that get to the bottom of the drivers of DOC export from forested headwaters are highly valuable to the readers of HESS.

*We really appreciate the constructive comments and the overall positive assessment of our study. Below we provide our responses to the reviewer's comments in italic font.*

**General (but still minor) comments:**

I suggest highlighting the importance of high-frequency data to identify "hot moments" of DOC export that would be covered if low-frequency data were looked at alone. A comparison of how many % of DOC were exported within only % of the time would be cool, from my perspective. Within your framework, you could highlight that X% DOC was exported during the autumn period, while this period only covers X% of the entire year.

*We agree that this information is interesting. However, only the flow-weighted values provide insight into mobilization processes as DOC is transported with water. We will insert a table in the SI that lists for each hydrological period the percentages of time, of runoff generated and of DOC export. We will also refer to this table in the results and discussion section and use the information provided there to support our interpretation.*

I suggest you make it more explicit that increasing concentrations with increasing discharge make DOC export during high flow periods disproportionally high. Stagnant or even decreasing concentrations could still show higher exports during high flow, but not as pronounced.

*We thank the reviewer for this suggestion and will express this point more explicitly in the first paragraph of chapter 4.1, before we discuss the flow-weighted DOC export values that disentangle the role of runoff generation for DOC export.*

What about stream density? Is it higher in the upstream catchments? I could imagine that a network of (temporary) streams largely increases hydrological connectivity. If, with your knowledge of the catchments, you agree – I suggest you add stream density and this argumentation.

*This is a good idea. However, we do not have data to include stream density in a quantitative way and would therefore prefer not to mention it.*

**Line-by-line comments:**

L9: "paramount importance to understand *all* processes of the global carbon cycle" sounds overstated to me; could you please tone it down a little?

> *We will change the sentence as follows: "...is of vital importance to understand the global carbon cycle in detail".*

L28-29: This doesn't read smoothly to me. If most of the carbon is inorganic, I miss the connection to organic carbon and the link to why organic carbon is also important. When you then, in L32-33, state that DOC is a major component of exported C, this appears contradictory to me.

> *We agree and will change the sentence in L 32 as follows: "However, especially in catchments with wetter and cooler..."*

Generally, I am missing at least a little information on the role of DOC for nutrient cycling (i.e., important electron donor for denitrification) and its impact on aquatic ecosystems.

> *Thank you for pointing this out. We will add following sentence to this section: "DOC is linked to nutrient cycling as it acts as electron donor in anaerobic respiration processes, e.g., denitrification (Lovley et al., 1999; Pang and Wang, 2021)."*

L60: have you looked at the TWI? It has often been found to correlate with DOC concentrations and could be interesting here as well. However, this is more a note than a comment that I want to see addressed.

> *For an earlier study, we calculated the TWI for the subcatchments and found that it did not differ much between the subcatchments. It was therefore, against first expectations, not helpful for explaining differences in DOC export and runoff generation behavior.*

L65: this section should be concretized. Why did you specifically want to go beyond the event scale (or low-frequency data)? What were your expectations? It reads a little listless.

> *We will adjust the section as follows: "The objective of this study was to improve our understanding of the temporal and spatial patterns of DOC export from a forested headwater catchment beyond the event timescale to assess the importance of seasons with differing hydrological conditions including low-flow periods for DOC export."*

L78: I assume HSsub should be HS$_{sub}$?

> *Yes, we will change it accordingly.*

L85: I would prefer you do not repeat exactly what is already given in the table. Either rephrase or take it out. And check this throughout the manuscript, please.

> *Thank you for this suggestion. We will rephrase this section as follows:  MG and KS represent*

the upper part of the catchment with  incised streams and $HS_{sub}$ represents the lower part of the catchment with a wide riparian zone. The geology of the Hinterer Schachtenbach catchment (HStot) is dominated by biotite granite  and cordierite–sillimanite gneiss .

*We will also check all other instances throughout the manuscript and make changes accordingly.*

L132: I very much appreciate this sound method section. However, I would like to see the fit between grab samples and sensor measurement as a scatterplot in the SI.

*We show this fit in Figures S3-S5 for the different devices.*

Table 3: I would like to have average discharge and DOC concentration included here as well to be able to tell if high loads are mainly due to high C or Q.

*We would like to not include this information in Table 3 as average values are not very meaningful, in our opinion. We rather refer to the new Table S1 in the SI where the relations of percentages of runoff and DOC export point to this aspect raised by the reviewer. Together with Figure 2 (time series) and Figure 4 (cumulative Q and DOC export for HStot), this demonstrates if high loads are due to high C, high Q or both.*

Figure 3 & Figure 6: I am a little confused with your unit of (flow-weighted) DOC export. According to your equation, DOC export is C [mg/L] * Q [L/min] * t [min], so the unit should be mg, right? How do you get mg /L*d in Figure 3 then and in Figure 6b mg/L...? What is it exactly that you define as flow-weighted DOC export, and how does it differ from DOC export and from concentrations? Please clarify.

*We either refer to absolute values of DOC export (kg) for a hydrological period with a certain length or to daily DOC export per day (kg/d). The daily values were derived by dividing the DOC export by the number of days of the specific hydrological period.*

*Flow-weighted DOC export is calculated as the total absolute DOC export value for a hydrological period (kg) divided by the cumulative absolute discharge of this period (L). By dividing this value by the number of days of the specific hydrological period, we get the daily flow-weighted DOC.*

*The concentration and the flow-weighted DOC export do have the same unit (weight/volume) but refer to two different aspects. Concentrations refer to a certain amount of C in the water column at a specific moment. DOC export is the total amount of DOC that was transported during a certain period, i.e. the load. Flow-weighted DOC export is the total amount of DOC that was transported during a certain period normalized by the total amount of water that was transported during the same period.*

Figure 5: Would it get too messy to add lines for cumulative Q here as well? Maybe as thin or transparent lines? I really liked being able to compare that to the export in the previous figure. But I leave that up to your discretion.

*We did consider a figure like that in the beginning but concluded that calculating cumulative values of discharge in 15 min resolution for HSsub (where we obtain values by calculating the differences between HStot on one hand and MG + KS on the other hand) is difficult. We would need to assume a certain time lag between the discharge values upstream*

*and downstream to account for the time it takes for water to travel from MG and KS to the outlet at HStot. However, this would be a very arbitrary decision, and therefore we decided to not do it.*

Figure 6a, b: What about the green (a) and black (b) columns reaching the limit of the y-axis? Are the values beyond the y-axis limit? Could you please change the limit or indicate these specific values somewhere in the figure?

> *We will modify the plots for a revised version of the manuscript, extending the y-axes and also removing the "all" columns.*

L51-52: The information in the last sentence could also very well be integrated into Table 4, which would be more consistent, from my point of view.

> *We assume that you refer to L251-252; we agree and will add this information to Table 4.*

L269-270: I would appreciate it if you could name these ranges briefly.

> *We will add the ranges to the manuscript:*
>
> *Agren et al., 2007: 14.8 to 99.1 kg ha$^{-1}$ yr$^{-1}$ (1490 to 9910 kg km$^{-2}$)*
> *Bernal & Sabater, 2012: 1.8 ± 1 kg C ha$^{-1}$ yr$^{-1}$ (1800 kg km$^{-2}$)*
> *Strohmeier et al., 2013: 84 kg C ha$^{-1}$ yr$^{-1}$ (8400 kg km$^{-2}$)*
> *As the exact values are not stated in Tittel et al., 2013, we will remove this source.*

L272-274: I partly disagree here. Precipitation is not simply equal to discharge... besides being driven by precipitation, discharge is further driven by catchment wetness that also relates to temperature controlling snowmelt and evapotranspiration, vegetation, and soil type – all of which control how much water is stored in the catchment and enable hydrological connectivity and transport. From your analysis and results, I would rather see a direct link to discharge than to precipitation alone.

Moreover, to me, "the most important factor" implies you have run some kind of statistic to rank the importance of factors.

Please rephrase your argument.

> *We agree that precipitation is not simply equal to discharge and that catchment wetness plays undoubtedly an important role, too. Depending on antecedent wetness and season, equally sized precipitation events may cause different runoff responses. However, in general, runoff events are caused by precipitation events (or snowmelt, which can be seen as an event as well), and this is what was meant here. We will rephrase the statements to avoid the impression of a simple relationship between precipitation and resulting discharge (beginning and end of section 4.1).*

L275: In your figures, you show the average daily DOC export, where can I see the absolute solute export? If it's a "new" result it should not appear here in the discussion for the first time.

> *The absolute DOC export can be found in Table 3 in the column "Total DOC Export (kg)".*

L298: Again, some numbers would help me here. What is the 'typical base flow concentration'?

> *We will add the numbers from Da Silva et al., 2021 (2.6 mg L$^{-1}$).*

L319: See my argument above. I agree that a lack of precipitation events can limit DOC flushing. However, especially in summer, there is not only a lack of precipitation but also higher evapotranspiration, reducing catchment wetness and connectivity and thus discharge.

*We will change the sentence as follows: "Moreover, it is possible that DOC had accumulated in the soils during summer due to 1) the warm temperatures, which enhance biological activity **as well as evapotranspiration**, and 2) the lack of precipitation events **and a low hydrological connectivity**, therefore limiting DOC flushing (Dawson et al., 2008; Kawasaki et al., 2005; Seybold et al., 2019; Wei et al., 2021).*

L320-326: I might be mistaken here. But is freshly fallen leaf litter directly turned into DOC? Doesn't it take some time to decay until it is **D**OC?

*According to Hongve (1999), fresh deciduous litter has very high potential for production of DOC in the short term. This study also observed a high leaching rate for fresh litter from leaf fall to early spring.*

L370: Good point! If not the entire area is connected, catchment area can be misleading.

L329: P ≠ Q; see my argumentation above

*We will adjust this section as follows: "Hence, precipitation regime and catchment wetness, both governing hydrological connectivity and runoff responses, were important factors for DOC export. Resulting runoff events contributed..."*

L410-414: You discuss this, and it sounds very reasonable to me, but you do not directly prove this. Thus, you should be careful with phrases like "is controlled by". Instead, "can be explained by", or  "we argue that..." would be more appropriate.

*We will tone down the statements as suggested. We will change the sentences mentioned as follows: "...hydrological connectivity, which is influenced by topographical position.... We argue that these hydrological processes control the DOC export...".*

L419-431: This is not really a conclusion, rather than an Outlook. Thus, I suggest calling this section "Conclusion and Outlook"

*We agree to changing the name of this section to "Conclusions and Outlook.*

---

## Author Comment (AC2)

**Authors' response to Comments of Reviewer 1**

**Review of hess-2024-250**

Title: Hydrological Controls on Temporal Contributions of Three Nested Forested Subcatchments to DOC Export, by Blaurock et al.

Blaurock et al. present high frequency DOC and discharge data for a period of one year from three nested forest headwater subcatchments in the Bavarian Forest National Park (Germany). They aim to explore differences in DOC export between the three subcatchments, which have distinct vegetation, microclimate, soil types, and topographical characteristics, among different "hydrological periods" namely spring, summer, autumn, winter, and snowmelt. Precipitation inputs drive overall exports whereas differences in runoff contribution and hydrological connectivity between the different catchments drive seasonal differences in DOC exports between the subcatchments.

High-frequency sensors and increasingly used to better understand biogeochemical mechanisms and mobilization processes at the catchment scale, particularly in relation to the important constituent DOC. This study provides further insights into the topic and should be of general interest for the readers of *Hydrology and Earth System Sciences*. I do have a number of concerns and suggestions and a list of other relatively minor comments that the authors should address before the manuscript can be accepted for publication.

*We thank the reviewer for their very constructive comments and the overall positive assessment of our study. Below we provide our responses to the reviewer's comments in italic font.*

**General comments**

The specific comments I provide below are extensive enough for the authors to consider the revision of their manuscript. Here I summarize my main points of criticism:

1. The definition of "hydrological periods" needs a more rigorous explanation, including e.g. information on how dry or rainy periods are determined.

   *Please see our reply to your comment "L 136-150". We agree that the definition of the hydrological periods needs to be explained in more detail.*

2. I miss a more compelling explanation/interpretation for the higher runoff generation at MG.

   *Please see our reply to your comment "L 351-362". We appreciate your constructive comments about this point.*

3. Likewise, I am not convinced about the explanations given for the low runoff generation and high flow-weighted DOC concentrations across all conditions observed at HSsub. The authors repeatedly mention that hydrological connectivity needs to be established at this site, but it appears that even during low hydrological connectivity HSsub provides water with high DOC concentrations. Where does it come from?

   *Please see our reply to your comment "L 375-380". We will expand the development of hypotheses that could explain the relatively lower runoff generation of $HS_{sub}$.*

**Specific comments**

Abstract

12. Maybe "soil type" instead of "soils"?

*We will replace "soils" by "soil types".*

12-14. In the abstract, you associate limited hydrological connectivity with snowmelt, summer, and winter. In general, snowmelt periods are associated with high hydrological connectivity instead, and I think it is similar in your study. Moreover, you do associate autumn with limited hydrological connectivity later on in the text (at least for subcatchments HSsub), so this part of the abstract is somewhat inconsistent with your interpretations.

*Indeed, the wording in the abstract is inconsistent with our main text. We will rephrase the respective sentences in the abstract.*

1 Introduction

33. More explicitly, "because of in-stream metabolic processes".

*This will be added to the text.*

35. I would say "In addition" rather than "But".

*We will correct this as suggested.*

44. The high groundwater levels also favour the build-up itself of organic matter in the soil (because of limited mineralization due to hypoxic conditions).

*We will add this point to the line of reasoning.*

47. I would say concentrations "generally" increase.

*We will add "generally" to the sentence, as suggested.*

2 Material and Methods

72. Do you perhaps mean "2.1 Study site"?

*Yes, thank you for pointing this out.*

75 – Figure 1. If I understand correctly, the study is based on data only from the Hinterer Schachtenbach catchment (delineated in red in the figure), which is part of the bigger Grosse Ohe catchment, which I understand is also depicted in the figure outside the delineated area. I would suggest to only present the Schachtenbach catchment in the figure, as the rest of the illustration is more distracting than informative.

*We will follow the suggestion of the reviewer and will show only the catchment Hinterer Schachtenbach with its subcatchments in the map in Fig. 1.*

90. Are these rock outcrops or exposed bedrock, or how are these rocks "interspersed" in the soils?

**Kommentiert [KB1]:** Ich denke, das reicht als Antwort, oder? Ich schaffe es mit QGiS leider nicht, aber kann es noch mit einem Bildbearbeitungsprogramm hinbekommen, denke ich.

**Kommentiert [KB2]:** Ich denke, das reicht als Antwort, oder? Ich habe es leider bisher nicht mit Qgis geschafft wegen Probeleme mit dem Koordinatensystem, aber notfalls schaffe ich es sicher mit einem Bildbearbeitungsprogramm und muss es halt manuell ausschneiden.

*The rocks (or small boulders) are mostly located within the soil profile. Some of them also intersect the soil surface and peak out. We will change the sentence to: "Almost 40 % of the area of MG is characterized by soils that are interspersed with rocks, mostly below the surface."*

91-95. Just out of curiosity: in the period 2018-2021 there was a drought followed by a large infestation of Norway spruce by bark beetles in large parts of Central Europe. Was the forest in your study are not affected by this disturbance?

*The forest in our study area (the "Rachel-Lusen area" within the Bavarian Forest National Park) was affected by large barkbeetle calamities in the 1990s and early 2000s. Since then, a stable and diverse forest has resulted from rejuvenation that can resist barkbeetle infestations better. However, other areas in the Bavarian Forest National Park have been hit by barkbeetle infestations in the past years.*

104-111. The gap filling at HS and Q construction at KS are fine. However, I am confused about the range of values shown in Figure S1 and Figure S2 compared to the range of values that you present here and that I could see in your raw data in the Figshare file. Specifically, the upper values are much larger in Figures S1 and S2 compared to the upper values shown in the study for all three subcatchments (e.g. the highest Q at HS according to the data here is 0.75 m3/s whereas it appears to be as high as ca. 3 m3/s in Figure S1 for an antecedent period). Could you clarify this point?

*The datasets used for gapfilling comprise data from before the study period that is presented in the manuscript. In Section 2.2.1 we wrote: "This data gap was filled using the discharge data of a neighboring catchment within the Große Ohe Catchment of similar size (Vorderer Schachtenbach, 5.9 km$^2$). Using the discharge values of one year prior to the data gap, a relationship between the discharge of the two catchments was established (R2=0.94) and calculated values were used to fill the data gap (Figure S1)." Also the tracer dilution experiments for discharge calculation in the subcatchment KS span a period that was not part of our study period (April-Dec 2021).*

*Figure S1 includes data from the snowmelt period of the year 2020, when a large snowmelt event led to discharge values as high as ca. 3 m$^3$/s in the beginning of February. This period is not presented in the manuscript, where we show the period June 2020 – May 2021. We will add the period from which data were taken for gapfilling to the figure caption in the SI.*

120-125. Thank you for this detail explanation. Just out of curiosity: the fact that you move D3 from KS to MG following the failure of D2 makes me think that you prioritized having data from MG compared to having data from KS. Is this correct and if so, why?

*Yes, that is correct. We had been working in the MG subcatchment since 2018, and our goal was to continue the data collection there to add to our multi-year dataset of MG. The measurements in the subcatchment KS started only with this study.*

125-135. Nice!

*Thank you!*

138-139. However, you have an additional period which you define as "snowmelt", which in fact is the longest of all.

*We will add following sentence: "In addition, we also introduced the hydrological period "snowmelt" to account for the very different hydrological conditions with increased runoff generation." (p. 6, L 144-145).*

139. Perhaps "large extent" instead of "larger extent".

*We will correct this as suggested.*

136-150. I think this classification is fine, but I wonder whether more details can be provided so it appears less arbitrary. For example, you mention "starts" or "ends" of "dry" or "rainy" periods to delimitate your hydrological periods, but no information on how you define a dry or rainy period is given or in reference to what. Also, the snowmelt period appears to be excessively long (Feb to Apr 2021). Did snow cover take that long to melt?

*We defined rainy and dry periods based on clear changes in precipitation regime and discharge response. For this, mean daily precipitation for preceding 14 day intervals, frequency of events and changes in discharge (lag times and peak values), together with visual inspection of the time series of precipitation and discharge, were considered. Admittedly, we did not perform statistical analyses but based our definition of hydrological periods on this more qualitative approach that was also supported by our combined experience and knowledge of the hydrologic response at the study site.*

*Due to the elevation difference between catchment outlet and catchment ridge (771-1373 m), the catchment is not snowfree until April or even early May. In 2021, there was a snow cover of 10 cm even at the lowest outlet of the catchment until the beginning of April. We do not have snow cover data for the upper part of the subcatchments but for adjacent catchments, where there was a snow cover of 15 cm present at the elevation of ca. 1300 m until the end of April. Snowmelt is also clearly visible as diurnal fluctuations in the discharge data.*

*In the revised version of the manuscript, we will add more detail to section 2.2.2 so that our definition of hydrological periods becomes clearer.*

168-170. Fair assumptions but what do you know about the in-stream processing of DOC in your system? I would be inclined to think that it is probably limited, but there is increasing evidence in the literature that in-stream DOC processing might be larger than previously thought, even in low order streams with non-labile DOC. What is the chemical character of the DOM? You can probably have a proxy for this with the absorbance data.

*The extent of in-stream processing of land-derived DOC is still not fully understood and a field of active research to better understand the export of terrestrial carbon into streams and its metabolic fate in aquatic systems. There are studies that indicate that the relevance of in-stream metabolic processes influencing stream DOC concentrations is limited (e.g., Singh et al. 2015 (https://doi.org/10.1002/hyp.10286); Bernal et al. 2019 (https://doi.org/10.3389/fenvs.2019.00060); Dawson et al., 2001 (https://doi.org/10.1016/S0048-9697(00)00656-2), at least in headwater streams where residence times of stream water are expected to be short. Very few studies have compared in-stream metabolic processing of DOC between periods of baseflow and event runoff, and it is not clear yet if runoff events stimulate or lower in-stream processing of DOC (e.g., Bernal et al 2019 (https://doi.org/10.3389/fenvs.2019.00060) and Demars 2018 (https://doi.org/10.1002/lno.11048).*

*DOM quality has been used to infer in-stream processing of terrestrial DOC. Kothawala et al. (2015, https://doi.org/10.1002/2015JG002946) found no indication for in-stream transformations to soil DOM composition, using absorbance metrics, in boreal headwater catchments.  A companion study to our research in the Hinterer Schachtenbach catchment investigated DOM composition of stream DOC and soil DOC sources and found: "The analyses revealed that at comparatively high dissolved organic carbon concentrations, the composition of DOM in-stream reflects the composition of DOM stored in the superficial soil layers", from which we concluded that there was no indication for alteration of DOC in the stream, i.e., no in-stream metabolic processing (da Silva et al. 2021 ([https://doi.org/10.1029/2021JG006425](https://doi.org/10.1029/2021JG006425))).*

*We will add the two latter references to this sentence to support our assumption that neglecting effects of in-stream metabolic processes on stream DOC concentrations are appropriate.*

3 Results

177-178. The "leading to sharply rising DOC…" implies that precipitation events are responsible for the increase. This is of course true, but via hydrological activation of upper soil layers that have build-up DOC during summer. Anyway, these explanations belong to the discussion so I guess what I am trying to say here is that you could avoid using terms like "leading" and simply describe the observed patterns without further implications on the processes.

*We agree with this comment and will change the sentence accordingly to: "In autumn, several large precipitation events occurred, and sharply rising DOC concentrations up to 20 mg/L were observed (Fig. 2)."*

191. Typically, DOC export is reported in kg/ha or g/m2. Please, transform the 3931 kg/km2 into either of these other units for better comparison with other studies.

*This will be done.*

210. Do you mean "DOC export" instead of "DOC concentration"?

*Yes, thanks for pointing this out.*

210-218. To me, the interesting thing about this kind of figure is to relatively compare the evolution of cumulative discharge and cumulative DOC export, with focus on when the lines deviate. For example, after both discharge and DOC evolved comparatively, DOC disproportionally increases at some point in mid-autumn, but this disproportionate increase is cancelled out during winter (with the exception of the mid-winter snowmelt event). Then again at the beginning of the snowmelt period DOC disproportionally increase relative to discharge, but as the snowmelt period advances, DOC decreases relative to the discharge, suggesting some kind of dilution effect or even production-limitation taking place then. You have some explanations in the discussion around this figure, but I would suggest to make these points more explicit, here in the results when you describe the figure, and later in the discussion.

*Thank you for this suggestion. We will slightly reword this paragraph and add the following sentence: "The deviation between the two cumulative curves indicated that DOC export increased disproportionately relative to discharge during these periods. In contrast, towards the end of the snowmelt period, DOC export decreased relative to discharge."*

229 – Figure 5. Again, I would rather present DOC exports in kg/ha or g/m2.

*We will insert revised versions of Figure 5 and Figure 6a, showing DOC export in kg/ha.*

242. But KS contributed less than expected according to its area ratio, right?

*In Figure 6c the relative contribution of the subcatchments to DOC export is shown and compared with the contribution that could be expected due to their area ratio. In the winter period, KS contributed more to DOC export than expected by area ratio (yellow bar and yellow line) whereas HS(sub) contributed less than expected (black bar and black line).*

4 Discussion

268. See my previous suggestions regarding the units of annual DOC export.

*We will correct the unit to kg ha$^{-1}$.*

312-313. I don't think I can agree here. The discharge time series show that flow is low towards the end of snowmelt, and therefore I can hardly imagine soils being saturated at this point. Also, according to Figure 4, DOC decreases with respect to discharge during this period, and it is only the activation of DOC source areas during spring rainfall events that can explain this pattern. Note as well that the way you define the different periods is very much influencing your findings in terms of DOC because you are using main DOC drivers in the definitions.

*We assume that after the end of snowmelt (the cited sentences refer to the beginning of the spring period) the soils are very wet and especially in the flat areas of the HS(sub) riparian zone saturated/close to saturation. Admittedly, the expression "high continuous DOC export" is not correct, as DOC export had decreased relative to discharge towards the end of the snowmelt period and picked up again during the first few weeks of the spring period. The sentence will be changed as follows: "Soils were saturated after the snowmelt period and, therefore, hydrological connectivity between the DOC sources and the stream existed facilitating the connection of distal DOC sources (Croghan et al., 2023) and an increased DOC export (Fig. 4) during the spring events."*

343-345. Also, potentially lower evapotranspiration as the deciduous trees drastically reduce transpiration during winter.

*Thank you for pointing this out. The sentence will read: "Additionally, litter fall (a) induces a lower water loss via canopy interception in autumn and (b) leads to strongly reduced transpiration rates. Both factors would result in shifts in the water balance, leading to an increased availability of water in the soil profile and active flow pathways, as suggested by the high runoff ratio at KS during winter (Table 4)."*

345. Rather than "after the snowmelt period", it could be better to write "during spring" "or from spring on".

*We will change the expression as suggested.*

351-362. To me, one of the most striking results in your study is the significantly higher runoff contribution of MG compared to the other two subcatchments, especially compared to KS which a priori are more comparable. In this part of the text you provide hypotheses that aim to explain this observation, but I am not convinced that they can fully account for it. What about the role of vegetation? After all, evapotranspiration is a main component of the water balance. I

can see that MG has the highest percentage of forest in a state of rejuvenation. Could it be that the overall evapotranspiration in this subcatchments is relatively lower than at KS because the forest is not fully developed and this contributes to the higher runoff contribution from MG?

*We will add this consideration of vegetation differences to this paragraph. We agree that this factor could also contribute to an increased runoff generation as it shifts the water balance. Ranking the hypotheses responsible for the increased runoff from MG, however, remains difficult. We would argue that the combination of the factors mentioned leads to the markedly higher runoff generation in MG as compared to KS. We will add following sentence: "The subcatchment MG also has the highest area percentage in the state of forest rejuvenation. It is therefore likely that due to these differences in extent of mature trees, the transpiration component in the water balance of MG is markedly lower compared to KS, resulting in an increased availability of water for runoff generation."*

375-380. But the higher flow-weighted DOC export at HSsub occurs across all periods and not only during high wetness. And, as you mention before in the text, autumn is a particularly limited period in terms of hydrologic connectivity between HSsub catchment soils and the stream, which "inhibit DOC mobilization". Therefore, I remain puzzled as to why HSsub (i) is so inefficient at generating stream runoff, especially during autumn (the runoff ratio of 0.13 is strikingly low), and (ii) can still provide water with high DOC concentration so that flow-weighted exports are high across all conditions. I think these points are the most critical to revise in a new version of the manuscript as they are also the most interesting in terms of catchment process understanding.

*We argue that the high flow-weighted DOC export in the subcatchment HSsub is the result of the interplay between hydrological processes and DOC production and accumulation. After the warm and dry summer with low runoff generation in the Hinterer Schachtenbach catchment, the soils in the flat riparian zone around the stream in HSsub had dried out, as did the microtopographic features ("ponds") in the riparian zone. The few events in August led to only small Q increases but still high DOC concentrations (resulting in a low DOC load). The month of September was particularly dry. Figure 2 in Blaurock et al. 2022 (https://doi.org/10.1029/2022JG006831) shows water levels below ground surface for three piezometers installed in the riparian zone of HSsub (one of them being installed in a pond) for summer and beginning of autumn 2020, thereby overlapping with the study period presented here. During July, August and September 2020, water levels in the soils did rise following precipitation events but also quickly receded again. This points to a quick drying-out of the soils in the riparian zone of HSsub and lost hydrologic connectivity. Particularly low water levels were reached at the beginning of August and at the end of September. The first autumn events led to a more substantial wet-up of the soils and produced strong DOC concentration increases in the stream (see Fig. 2). However, Q remained still low. The highest value of water level in P1 (again Fig. 2 from Blaurock et al. 2022) corresponds to spill-over of the pond; this is the occasion when the ponds are fully filled, flow over and connect with each other and the stream. This spill-over was reached only once during June to end of October 2020, and even though the pond remained filled during most of October (water level > 0 cm), it did not spill again. This indicates that wide-spread hydrological connectivity was not reached yet, therefore runoff generation in HSsub remained still low. Towards the end of the autumn period, in November 2020, the baseflow level of Q rose, suggesting that runoff generation had picked up. Unfortunately, we stopped our piezometer measurements at the end of October 2020.*

*This dependence of the establishment of hydrologic connectivity on the flat topography in the riparian zone of HSsub is paired with the high accumulation of DOC in the soils with often higher groundwater tables (which limits mineralization of DOC) and the presence of the microtopographic features, the ponds. Our analyses presented in Blaurock et al. 2022 showed that upon spill-over and connection of the ponds the DOC signature of the pond DOC can be found in stream DOC during event flow. At the same time, the pond DOC concentrations can rise up to values of 100 mg/L (unpublished data, observed during a field hydrology course at the same site in 2023), which points to a very strong DOC pool being present in these ponds in the riparian zone of HSsub.*

*In a revised version of the manuscript, we will explain in a little more detail the role of the ponds as DOC accumulation and release hot spots and also refer to the Figure 2 in the Blaurock et al. 2022 study in chapter 4.2.*

382. In Figure S6, how is it that the mean of daily flow-weighted DOC export of "all" periods is lower than each of the periods for all three subcatchments? Are you using the summer period to calculate the mean in "all"? Even if you do, I find it difficult to arrive to those values. I would expect to see something similar in relative terms to what is shown in Figure 6b (in fact, the values should be proportional so the relative differences should be the same).

*The flow-weighted values for "all" are in a similar range as for the individual hydrological seasons (because they refer to a larger amount of Q). When this value is divided by all days (entire study period), the resulting value becomes small. However, after evaluating the "all" columns" again, we decided to remove them in a revised version of this figure (and Figure S6). We realized that we do not refer to the "all" values in the text because the focus is on comparing the hydrological periods. Therefore, the "all" columns are confusing. Thank you for pointing this out.*

5 Conclusions

425-431. Droughts could also lead to bark beetle infestation and death of trees, with important hydrological and biogeochemical consequences at the catchment scale.

*We agree that droughts can lead to the death of trees, e.g., by facilitating bark beetle infestations and/or windthrow. This would affect the partitioning of water within the catchment water balance and provide a disproportionately high carbon pool for decomposition within a short period. In a revised version of the manuscript we will incorporate this thought into the text of the conclusion section.*

---

## Author Response (AR2)

**Point-by-point reply to comments on HESS manuscript HESS-2024-250.R1**

**"Hydrological Controls on Temporal Contributions of Three Nested Forested Subcatchments to DOC Export"**

*By Blaurock, Beudert and Hopp*

*Dear Editor,*

*Below, you find our point-by-point reply to the remaining comments by the reviewers. We thank you and the reviewers once again for the overall positive assessment of our study and the very constructive review comments.*

*With kind regards,*

*Luisa Hopp (corresponding author) on behalf of all authors involved*

**Authors' response to Comments of Reviewer 2**

Referee #2:

In their revised manuscript, Blaurock et al. have incorporated my comments to my full satisfaction or explained comprehensibly why they prefer not to do so. I have only one minor remark left, which you can find below. With this remark being addressed, I am convinced that this manuscript will be a very valuable contribution for the readers of HESS.

L 145-147: I missed this in the first round, but could you please explain how you identified snowmelt periods? Via temperature? Could you please add this to your method section?

> *We explain how we defined snowmelt periods in L 152 – 154: "Snowmelt was defined as the period when temperatures rose above zero, a snow cover was present, and snowmelt was clearly visible in discharge data at HS as diurnal fluctuations. The snowmelt period ended when the catchments were completely free of snow at all elevations."*

**Authors' response to Comments of Reviewer 1**

Referee #1:

Blaurock et al. submitted a revised version of their manuscript, along with their replies to the reviewer comments. I have carefully reviewed both the replies and the revised manuscript, and I thank the authors for their engagement and for making the pertinent changes to the manuscript. I am generally satisfied with the revisions made, though not entirely. I believe this version of the manuscript is close to acceptance; however, some minor (yet important) improvements remain to be addressed, as outlined below.

Remaining comments

L. 14-15. I am not satisfied with the rephrasing of this sentence. "[...] due to the interplay between runoff generated and DOC available for mobilization" is vague/ambiguous. Perhaps "[...] due to relatively lower DOC available for mobilization and runoff generation".

*We changed the sentence to: "...DOC export was low due to either low runoff generated and/or relatively lower DOC available for mobilization."*

L. 69-70. I did not mention it before, but this sentence misses a mention to the study site, e.g. "In an earlier study in a temperate forest headwater catchment".

*We inserted the addition as suggested.*

L. 92. "Hinterer Schachtenbach catchment" not needed as it has already been defined, just say "HStot".

*Accepted.*

L. 182. Thank you for adding this information but maybe instead write "[...] in similar forested headwater streams", or "[...] in other cold-climate forested headwater streams", otherwise the sentence is too strong (and likely untrue).

*We inserted "similar".*

L. 189-190. Grammatically odd. Please, remove "and" and "were observed" from this sentence.

*We rephrased the sentence.*

L. 228-229. And during most of the winter.

*We rephrased this sentence.*

L. 314. I would say "DOC produced during summer and autumn [...]".

L. 326-328. I am still not convinced by this reasoning. First, soils might have been wet, but I can hardly believe that they were saturated. You do not provide observations supporting this so you also need to be careful about wordings such as "soils were" and instead using wordings such as "soils likely were". This "wetness" state might facilitate hydrological connectivity, but only during rainfall events that could activate DOC source areas. The way you formulate the sentence it implies that hydrological connectivity between DOC sources and stream were constant during the spring period, which I don't think is the case.

*We rephrased the sentence as follows: "Soils were likely saturated after the snowmelt period and, therefore, hydrological connectivity between the DOC sources and the stream facilitated the connection of distal DOC sources (Croghan et al., 2023). This connection contributed to an increased DOC export (Fig. 4) during the spring events."*

L. 352. Vague. Please, be more specific.

*The section the referee refers to is the conclusion of the first chapter of the discussion, summarizing in general statements our specific observations and interpretations above.*

L. 373-374. This might be a factor, but if true it would not imply that MG is more efficiently at generating runoff, only that you have not accounted for all the water inputs in this catchment. Please, make this point explicit.

*In this section, we are developing some possible reasons for the disproportionately higher runoff generation in the subcatchment MG, as compared to the neighboring*

*subcatchment KS. We are not stating anywhere in the text that MG is more efficient in runoff production. We would prefer to keep this section as it is.*

L. 393-418. As I mentioned before, I was puzzled as to why HSsub (i) was so inefficient at generating stream runoff, especially during autumn (the runoff ratio of 0.13 is strikingly low), and (ii) could still provide water with high DOC concentration so that flow-weighted exports are high across all conditions. You made revisions in this part of the text to address my concern and the explanations are closer to what I would consider satisfactory, but further clarifications are needed. You propose the ponds present at HSsub as the potential source of large DOC influx to the stream from this subcatchment. And you describe that they "fill regularly with water from the bottom during large events and periods of high catchment wetness". Yet, at the same time you argue that summer and also autumn are periods where soils dry out and hydrologic connectivity is "lost". **So how can ponds be effective sources of DOC to the stream during autumn if the conditions for its hydrological connection to the stream are not met?** I see some contradictions in the reasoning. Perhaps there is a different way in which these ponds can be hydrologically connected, e.g. via "overland" flow during rainfall events even during low wetness? Or is it that the rainfall events during autumn can quickly connect upper DOC-rich riparian layers even in a context of low wetness and low overall hydrological connectivity? (some kind of "ephemeral" establishment of hydrological connectivity with DOC sources?). Please, give further thoughts to the processes that can explain the observations and produce a more coherent narrative about this particular point.

*In our opinion, we discuss most of the points the referee mentions already in this section. The ponds in the riparian zone can be sources of DOC also during autumn because our observations and measurements have shown how they rapidly fill and empty again during and following late summer/early autumn rainfall events. We describe this in L 408-420, based on data that we presented in an earlier article. Based on the suggestions of the referee, we made some additions to the text that hopefully make our reasoning even clearer.*